# Enteric Pathogens in Wild Boars Across the European Union: Prevalence and Antimicrobial Resistance Within a One Health Framework

**DOI:** 10.3390/antibiotics14121246

**Published:** 2025-12-10

**Authors:** Francesca Piras, Giuliana Siddi, Enrico Pietro Luigi De Santis, Christian Scarano

**Affiliations:** Department of Veterinary Medicine, University of Sassari, Via Vienna, 2, 07100 Sassari, Italy; g.siddi1@phd.uniss.it (G.S.); desantis@uniss.it (E.P.L.D.S.); scarano@uniss.it (C.S.)

**Keywords:** *Salmonella*, *Yersinia enterocolitica*, *Campylobacter*, Shiga-toxins producing *E. coli*, wildlife sentinels, human–livestock–wildlife interface, resistome, integrated surveillance

## Abstract

Wild boars, widely distributed across natural, agricultural, and urban landscapes, represent an ideal sentinel species for monitoring the emergence and spread of antimicrobial resistance (AMR) at the human–wildlife–livestock interface within the One Health framework. This review summarizes current knowledge on the prevalence, diversity, AMR, and epidemiological significance of major enteric pathogens isolated from wild boars in the European Union, with particular attention to their potential role in AMR dissemination. Numerous studies have reported variable prevalence rates for *Salmonella* spp., *Yersinia enterocolitica*, Shiga toxin-producing *Escherichia coli* (STEC), and *Campylobacter* spp. High prevalence rates has been observed in fecal samples—35% for *Salmonella*, 27% for *Y. enterocolitica* and STEC, and 66% for *Campylobacter*—highlighting the role of wild boars as carriers and the associated risk of carcass contamination during slaughter. Tonsils represent a key niche for *Y. enterocolitica*, with prevalence reaching 35%. Several studies have identified resistance to antimicrobials classified by the World Health Organization as critically important or high priority for human medicine, including fluoroquinolone-resistant non-typhoidal *Salmonella* spp. and third-generation cephalosporin-resistant *Y. enterocolitica*, raising notable public health concerns. Despite increasing interest, most available studies remain descriptive and geographically limited, providing limited insight into AMR acquisition and transmission pathways in wild boars. New approaches—such as resistome analyses and epidemiological cut-off values—offer added value to distinguish wild-type from acquired-resistant strains and to better understand AMR dissemination dynamics. Integrating wildlife into One Health surveillance systems is essential to capture the full complexity of AMR spread.

## 1. Introduction

Wild boar (*Sus scrofa*) is one of the most widely distributed mammals worldwide, with populations increasing steadily over the past 30 years [1]. This growth is driven by its *r*-strategist reproductive traits [2,3] and marked ecological adaptability, as an opportunistic omnivore capable of adjusting its diet and thriving in diverse environments—from natural and agricultural areas to peri-urban and urban settings—while exploiting anthropogenic resources [4,5].

Other factors contributing to the increase in wild boar numbers include the scarcity of natural predators: humans represent the main cause of mortality, primarily through hunting and, to a lesser extent, traffic collisions [4]. Additionally, habitat fragmentation can influence wild boar movement and distribution, leading to increased mobility in search of resources, altered foraging behavior, and potentially greater human–wildlife conflicts [6,7].

As a consequence of population growth and geographic expansion, new challenges have emerged. The broad distribution of wild boars and their presence across diverse habitats highlight their major epidemiological role as hosts and reservoirs for several zoonotic agents such as *Mycobacterium bovis* [8], *Brucella* spp. [9], and pathogens capable of spill-over to domestic animals, such as African Swine fever virus [10]. There is a growing body of scientific literature on the role of wild boars as carriers of pathogens and indicator bacteria that may also harbor antimicrobial resistance determinants, as *Salmonella* spp., Shiga-toxins producing *E. coli* (STEC), *Yersinia enterocolitica*, *Campylobacter* spp., methicillin-resistant *Staphylococcus aureus* (MRSA), vancomycin-resistant enterococci, etc. [11,12,13,14,15,16].

Moreover, the synurbization—a specific form of synanthropization defined as the adaptation of animal populations to human-modified environments—of wild boars increases the likelihood of pathogen spillover events. Their frequent interactions with domestic animals, pets, and humans in peri-urban and urban environments facilitate the transmission of enteric pathogens and antimicrobial-resistant bacteria across ecological boundaries [13].

Antimicrobial resistance (AMR) is a major global challenge for both human and animal health, accelerated by human activities [17,18]. Many antimicrobials used in veterinary medicine and agriculture belong to the same classes as those used in human medicine [19], including substances deemed “critically important” by the World Health Organization [20]. Managing AMR therefore requires a coordinated One Health approach [21], which acknowledges the interconnectedness of human, animal, plant, and environmental health [22].

Recently, the One Health model has been further refined to include the pivotal role of the geographical proximity of ecosystems in the emergence and dissemination of health-related traits [23]. Within this framework, AMR represents the quintessential One Health issue [24]. Bacteria and their genes have a remarkable capacity to adapt and mutate rapidly, and to move within and between humans, animals, and the environment through horizontal gene transfer [25]. AMR (particularly in Enterobacterales at the fecal level) has been already widely observed in wild reptiles, birds, and mammals [26,27,28,29]. However, knowledge of the transmission dynamics within environmental and wildlife compartments remains limited, and the directionality of these mechanisms is complex to determine and mostly assumed [5]. In this context, wild boars represent the ideal model species for understanding the emergence, spread, and persistence of AMR at the human–wildlife–livestock interface.

The aim of this comprehensive review was to analyze current knowledge on the prevalence, diversity (in terms of identified species, serotypes, and biotypes), AMR, and epidemiological significance of major enteric pathogens (*Salmonella*, *Y. enterocolitica*, STEC, and *Campylobacter*) isolated from wild boars in the European Union. Novel approaches, such as resistome analysis and the use of epidemiological cut-off values, were also considered, as they help address existing gaps in the understanding of AMR acquisition mechanisms and transmission routes.

In addition, the review examined the role of the wild boar as a model of synurbization, highlighting how the increasing expansion of this species into peri-urban and urban environments enhances opportunities for interactions with humans, domestic animals, and anthropogenic sources of contamination. This aspect is particularly relevant when considering sentinel bacteria for AMR surveillance, such as *Enterococcus* spp., which can reflect environmental and human-associated antimicrobial pressure.

## 2. Materials and Methods

This comprehensive review focused on the four leading zoonotic agents in the European Union, as identified in the most recent EFSA and ECDC report [30]: *Campylobacter* spp., *Salmonella* spp., Shiga toxin-producing *E. coli* (STEC), and *Y. enterocolitica*.

A structured literature search was conducted using the SCOPUS database. The search strategy included combinations of keywords such as “wild boar”, “*Sus scrofa*”, “antimicrobial resistance”, “AMR”, “antibiotic resistance”, “*Salmonella*”, “*Yersinia enterocolitica*”, “STEC”, “Shiga toxin–producing *Escherichia coli*”, “*Campylobacter*”, “Enterobacterales”, “foodborne pathogens”, and “Europe”. Moreover, studies regarding *Enterococcus* spp. were also included as considered important sentinel bacteria for antibiotic resistance surveillance.

The review includes studies published over the past 15 years to provide a comprehensive overview of the prevalence and antimicrobial resistance of enteric pathogens isolated from wild boars. Although this extended time span allows the inclusion of a larger body of literature, it is acknowledged that diagnostic methods, antimicrobial susceptibility testing standards, and breakpoint interpretations have changed over the years, complicating the interpretation of temporal trends. The primary focus of the review remains on more recent studies (last 10 years), whereas older studies are included to provide historical context and to illustrate long-term patterns.

Titles and abstracts were screened to identify studies reporting occurrence, phenotypic and/or genotypic antimicrobial resistance in the target microorganisms isolated from wild boars.

Exclusion criteria included studies conducted outside the European Union.

## 3. Silent Carriers at the Human–Wildlife Interface: Enteric Pathogens in European Wild Boars

The scientific literature provides substantial evidence of the role of wild boars as reservoir of enteric pathogenic microorganisms, with particular emphasis on the four leading zoonotic agents in the European Union, as identified in the most recent EFSA and ECDC report [30]: *Campylobacter* spp., *Salmonella* spp., STEC and *Y. enterocolitica*.

Investigations into the carrier status of wild boar have been conducted primarily on fecal samples, although other matrices such as lymph nodes and tonsils are also commonly examined. In addition, to evaluate the risk of meat contamination during slaughter and subsequent processing, analyses frequently include carcass, meat and organs surface, such as the liver.

In the following sections, findings on prevalence, influencing factors, and diversity of *Salmonella* spp., *Y. enterocolitica*, *Campylobacter* spp., and STEC from studies conducted in various European Union countries are presented.

### 3.1. Prevalence and Diversity of Salmonella spp.

*Salmonella* spp. is a major cause of food-borne illness in the world, with an incidence of 18.0 cases per 100,000 inhabitants in the European Union [30] and an estimate of 1.35 million infections in the United States every year [31].

Domesticated animals are considered the main reservoir of non-typhoidal *Salmonella*, spp. but wild species can also act as asymptomatic carriers, and their role has gained increasing attention in recent years [32].

Carriage of *Salmonella* spp. in wild boars has been investigated in several studies across different European Union countries.

Table 1 summarizes the prevalence of *Salmonella* spp. detected in different sample types (feces, lymph nodes, carcass surface, skin, tonsils, organs, blood) collected from wild boars in various European Union countries. The sampling period is indicated. Prevalence is expressed as the number of positive samples out of the total number examined, with the corresponding percentage reported in brackets. Details regarding the detection method (culture, molecular, and/or immunoenzymatic assays) are also provided.

Reported prevalence in fecal samples shows wide variability, ranging from 0% to 35.6%. In lymph nodes, prevalence rates vary between 0% and 17.8% [11,13,33,34,35,36,37,38,39,40,41,42,43,44,45,46,47,48,49,50]. Comparisons between studies are not always straightforward, as testing procedures vary. Differences in analytical methods (culture vs. culture combined with PCR) and sample types (intestinal contents vs. rectal swabs) can markedly influence estimates of *Salmonella* spp. prevalence, with rectal swabs generally less sensitive [13,33,40,43]. Fecal shedding is also intermittent in carrier wild boars, particularly after the acute phase, reducing detection probability [51,52]. Additionally, the amount of feces analyzed (1–25 g) significantly affects diagnostic sensitivity, as shown in both wild boar studies and pig research [33,36,42,53,54,55].

The fecal–oral route is considered the most frequent pathway for *Salmonella* spp. infection, which is why mesenteric lymph nodes are commonly examined. However, inhalation of contaminated aerosols or dust and infection via the respiratory tract are alternative routes [56,57]. Consequently, tonsils and other oral cavity tissues, which drain to the sub-mandibular lymph nodes, should also be included in studies. Sannö et al. [40] reported a *Salmonella* spp. prevalence of 12% in wild boar sub-mandibular lymph nodes, slightly higher than the 10% found in mesenteric lymph nodes but lower than the 14.7% observed in tonsils. Likewise, Gil Molino et al. [43] found *Salmonella* spp. prevalence to be approximately three times higher in tonsils (18.7%) than in mandibular lymph nodes (5.1%), underlining the importance of including these samples when assessing *Salmonella* spp. circulation in wild boar.

Several interconnected factors influence *Salmonella* spp. prevalence in wild boar populations, including habitat, climate, population structure, and sampling season [42,58]. Age is particularly important: younger animals (<24 months) show higher prevalence due to immature immunity, unstable gut microbiota, and greater exposure [36,38,46]. Age also affects detection by sampling site, as piglets tend to carry *Salmonella* spp. in lymph nodes, whereas older pigs shed it in feces, enhancing transmission [58]. Gender and seasonality further contribute, with higher prevalence in socially grouped females [33] and during warmer months, when environmental conditions promote bacterial survival and fecal–oral spread [41,42].

In the studies examined, a high variability of *Salmonella* spp. serovars and/or subspecies has frequently been reported. For instance, Chiari et al. [36] identified 30 different serotypes classified in three subspecies—*enterica*, *diarizonae* and *houtenae*—and Zottola et al. [38] identified 15 different serovars of *S. enterica subsp. enterica*, along with more than 20 isolates belonging to other subspecies such as *salamae*, *diarizonae*, and *houtenae*. Similarly, eight different serovars were detected in a wild boar population inhabiting the Asinara Natural Park, a small island off the northern coast of Sardinia with a surface area of only 51.9 km^2^, further highlighting the remarkable diversity of serovars [46].

This pattern is commonly observed in wildlife in general, and in wild boars in particular, as a result of their multiple sources of exposure, including livestock farming and waste disposal [32,36]. In addition, the omnivorous feeding habits of wild boars contribute to this variability, since they may consume other mammals, birds, reptiles, and amphibians that act as *Salmonella* spp. carriers [43,59].

A point of concern is that some of the most frequently detected serovars in wild boars—such as Typhimurium, Coeln, Enteritidis, Thompson, and Newport [38,42,46]—are among the most commonly reported in human salmonellosis cases in Europe in 2023 [30]. Another notable finding is the detection of serovars shared with livestock in areas characterized by extensive cattle farming, suggesting a possible spillover between the two species [60]. Conversely, other authors [42], using Pulsed-Field Gel Electrophoresis (PFGE), compared isolates from wild boars with those recovered from domestic pigs in an area characterized by intensive farming, finding an overlap only at the serotype level and none at the PFGE level, thus indicating the effectiveness of the biosecurity measures implemented in this type of farming system.

Taken together, these findings highlight the relevance of wild boars as a key species for understanding *Salmonella* circulation at the human–animal–environment interface, underscoring the importance of adopting a One Health perspective in surveillance and risk assessment.

Figure 1 and Figure 2 show the number of *Salmonella enterica* subspecies and the number serotypes of *Salmonella enterica* subs. *enterica* identified, respectively.

**Table 1 antibiotics-14-01246-t001:** *Salmonella* spp. prevalence expressed as positive/total (%) investigated in different sample types collected from wild boars in various countries of the European Union.

Country	Sampling Period	Source	Positive/Total (%)	Comments on Detection Method	Ref.
Portugal	2005–2006	Feces (rectum content)	17/77 (22.1)	Culture	[34]
Switzerland	2007–2008	Feces (not specified)	0/73	RT–PCR	[33]
			0/73	Culture (for RT-PCR positive samples)	
		Tonsils	19/153 (35.8)	RT-PCR	
			8/153 (5.2)	Culture (for RT-PCR positive samples)	
Italy	2007–2010	Feces (intestinal content)	326/1313 (24.8)	Culture	[36]
Spain	2007–2011	Feces (rectum content)	66/214 (30.8)	Culture	[35]
Spain	2009–2011	Feces (rectum content)	1/574 (0.17)	Culture	[37]
		Carcass surface	5/585 (0.8)		
Italy	2010–2012	Feces (colon content)	54/499 (10.8)	Culture	[38]
		Blood (serum)	255/383 (66.5)	Enzyme-Linked Immunosorbent Assay	
Spain	2010–2015	Feces (colon content)	25/838 (2.9)	Culture	[43]
		Lymph nodes (sub-mandibular)	21/415 (5.1)		
		Tonsils	40/214 (18.7)		
Italy	2012–2013	Muscle (diaphragm or leg)	7/194 (3.6)	Enzyme-Linked Fluorescent Assay followed by culture for positive samples	[61]
Portugal	2013–2014	Feces (rectum content)	1/21 (4.7)	Culture	[39]
Serbia	2013–2014	Feces (rectum content)	13/425 (3.1)	Culture	[50]
		Carcass surface	4/425 (0.9)		
		Skin surface	1/425 (0.2)		
Italy	2013–2017	Liver	260/4335 (6)	PCR followed by culture for positive samples (results referred to cultural positivity)	[62]
Sweden	2014–2016	Feces (not specified)	7/90 (7.8)	RT-PCR preceded by culture (results referred to PCR positivity)	[40]
		Lymph nodes (mesenteric)	9/90 (10)		
		Lymph nodes (sub-mandibular)	3/25 (12)		
		Tonsils	20/136 (14.7)		
Denmark	2014–2016	Feces (rectum content)	0/115	Culture	[44]
Spain	2015–2016	Feces (rectal swabs)	4/130 (3.1)	Culture	[13]
Serbia	2015	Carcass surface Skin surface	4/210 (1.9) 3/210 (1.4)	Culture	[63]
Germany	2016	Feces (not specified)	13/552 (2.4)	Culture	[47]
Finland	2016	Blood (serum)	69/181 (38)	Enzyme-Linked Immunosorbent Assay	[64]
		Organs (kidney and spleen)	6/130 (4.6)	RT-PCR followed by culture for positive samples (results referred to PCR positivity)	
Italy	2016–2017	Feces (caecum content)	4/57 (7)	Culture	[41]
		Lymph nodes (mesenteric)	2/57 (3.5)		
		Carcass surface	0/30		
Italy	2016–2019	Feces (colon content)	32/90 (35.6)	Culture	[46]
		Lymph nodes (mesenteric)	16/90 (17.8)		
		Carcass surface	1/90 (1.1)		
Italy	2017–2018	Feces (not specified)	30/189 (15.9)	Culture	[42]
		Lymph nodes (mesenteric)	6/189 (3.2)		
Italy	2017–2020	Feces (not specified)	Not reported	Culture (previous study)	[48]
		Lymph nodes (mesenteric)			
		Carcass surface			
Italy	2018–2019	Carcass (excision method)	3/120 (2.5)	Culture	[65]
Italy	2018–2023	Feces (rectum content)	1/280 (0.3)	Culture	[49]
		Carcass surface	5/280 (1.8)	Enzyme-Linked Fluorescent Assay	
		Liver	3/280 (1.1)	Enzyme-Linked Fluorescent Assay	
Italy	2018–2020	Feces (rectal swabs)	7/287 (2.4)	Culture	[45]
		Spleen	6/287 (2)		
		Liver	5/287 (1.7)		
Italy	2019	Tonsils	0/36	Culture	[66]
		Carcass surface	0/36		
		Meat (forearm area)	1/36 (2.8)		
Italy	2020	Carcass surface	5/64 (7.8)	Culture	[67]
		Lymph nodes (mesenteric)	5/64 (7.8)		
Italy	2020–2022	Feces (colon content)	3/66 (4.5)	Culture	[11]
		Lymph nodes (mesenteric)	0/66		
		Carcass surface	0/49		

RT-PCR: Real-Time Polymerase Chain Reaction.

### 3.2. Unveiling Y. enterocolitica: Prevalence Trends and Bio-Serotypes

Between the microorganisms of the genus *Yersinia*, *Y. enterocolitica* is the species most frequently linked to human disease (yersiniosis). *Y. enterocolitica* is classified into six biotypes (BT) with varying pathogenicity: 1A (non-pathogenic), 1B (highly pathogenic), and 2–5 (weakly pathogenic). It is further subdivided into over 70 serotypes based on antigenic differences. The main bio-serotypes responsible for human yersiniosis are 1B/O:8, 2/O:9, 2/O:5,27, and 4/O:3 [68,69].

Yersiniosis is currently the fourth most common zoonosis in Europe, with a number of cases of 8738 in 2023 [30]. The pig is considered the main reservoir of *Y. enterocolitica*, from which strains belonging to pathogenic bio-serotypes for humans are most frequently isolated, while reports on the occurrence of *Y. enterocolitica* in wild boars are scarcer, and the epidemiological link with domestic pigs is still unknown [70].

As regards pigs, the tonsils represent the ecological niche of the microorganism [71,72] and this source is usually analyzed also in wild boars’ investigations.

Table 2 summarizes the prevalence of *Y. enterocolitica* detected in different sample types (feces, lymph nodes, carcass surface, meat, tonsils, organs, blood) collected from wild boars in various European Union countries. The sampling period is indicated. Prevalence is expressed as the number of positive samples out of the total number examined, with the corresponding percentage reported in brackets. Details regarding the detection method are also provided: culture (cold and/or warm enrichment) and/or molecular or immunoenzymatic assays.

Comparison between studies is not always straightforward. In addition to the factors already mentioned for *Salmonella* spp.—such as sample size-variability is also influenced by the different cultural methods employed. In fact, warm and/or cold cultures are often applied in parallel to exploit the uncommon ability of this enteric pathogen to replicate at refrigeration temperatures [69]. Cold enrichment remains more sensitive and reliable for detecting *Y. enterocolitica* in animal matrices, though it is slower, as demonstrated by different investigations carried out in wild boars [73,74,75]. Warm enrichment may be useful for rapid screening, but combining both methods increases overall detection probability and strain diversity [73,74,75].

As in domestic pigs, the tonsils appear to act as the main niche for the microorganism, with prevalence rates as high as 35% reported in Switzerland [33] and 17.9% in Germany [70]. In the latter study [70], 89.5% of isolates were identified as BT 1A, while the remaining belonged to BT 1B. Although BT 1B is generally considered highly pathogenic for humans, in this case the isolates lacked the virulence plasmid, raising questions about their pathogenic potential. In the same study [70], the highest prevalence was observed in animals aged 12–24 months, although the influence of age was not statistically evaluated. Multi Locus Sequence Typing (MLST) analysis revealed a high degree of heterogeneity among BT 1A isolates, despite samples being collected within a small hunting area during a single hunting season.

In fecal samples, prevalence ranges between 0% and 27% [11,33,67,75,76]. As in the tonsils, most isolates were identified as BT 1A, but other BTs, including serotypes 2/O:9 and 4/O:3—commonly associated with human yersiniosis and pigs—were also detected, albeit in low numbers. Syczylo et al. [74] in Poland identified BT 1B in isolates from rectal swabs samples. These isolates only harbored the *yst*B gene, which is associated with strains of low or uncertain pathogenic potential, but its presence suggests the capability to induce gastrointestinal symptoms, especially in susceptible subjects [77].

Regarding lymph nodes, prevalence rates of 5–6% were found in the mesenteric district [13,40]. Interestingly, Sanno et al. [40] reported a higher prevalence (12%) in submandibular lymph nodes, similar to that detected in tonsils (14%), presumably due to the anatomical proximity between these sites.

Several studies have reported seasonal variations, with higher prevalence rates observed in autumn and winter, coinciding with the wild boar hunting season in most European countries. This trend is likely related to stress factors such as low temperatures and food scarcity [13,78,79], but also to the already mentioned ability of *Y. enterocolitica* to multiplicate at low temperatures.

Interestingly, the presence of high-density ovine populations has also been identified as a factor influencing *Y. enterocolitica* prevalence in wild boars, as reported by Arrausi-Subiza et al. [78]. Sheep have been described as reservoirs of *Y. enterocolitica* [80,81], and sheep milk has also been identified as a source of potentially pathogenic strains [82], highlighting once again the relevance of a One Health perspective in understanding pathogen transmission dynamics at the wildlife–livestock–environment interface.

**Table 2 antibiotics-14-01246-t002:** *Y. enterocolitica* prevalence expressed as positive/total (%) in different sample types collected from wild boars in various countries of the European Union.

Country	Sampling Period	Source	Pos/Tot (%)	Comments on Detection Method	Ref.
Switzerland	2007–2008	Feces (not specified)	4/73 (5.5)	RT-PCR	[33]
			1/73 (1.3)	Culture (direct plating of RT-PCR positive samples)	
		Tonsils	26/73 (36)	Real-Time PCR	
			6/73 (8)	Culture (direct plating of RT-PCR positive samples)	
Italy	2008–2010	Carcass surface	3/251 (1.2)	Culture (cold enrichment)	[83]
Spain	2009–2012	Tonsils	24/72 (33.3)	RT-PCR followed by cultural (direct plating of RT-PCR enrichment of RT-PCR positive samples)	[78]
Poland	2012–2013	Feces (rectal swabs)	40/151 (26.5)	Culture (warm and cold enrichment)	[75]
Germany	2012–2013	Tonsils	19/111 (17.1)	Culture (warm enrichment)	[70]
Italy	2012–2013	Muscle (diaphragm and leg)	34/230 (14.8)	Culture (cold enrichment)	[61]
Czech Republic	2013–2014	Meat juice from diaphragm	89/135 (81.9)	Enzyme-Linked Immunosorbent Assay (antibodies anti-*Yersinia* spp.)	[84]
Poland	2013–2014	Feces (rectal swabs)	110/434 (25.3)	Culture (warm and cold enrichment)	[74]
Italy	2013–2018	Liver	126/4890 (2.6)	Culture (warm enrichment)	[85]
Italy	2014–2015	Muscle (*Longissimus dorsi*)	0/22	Culture (cold enrichment)	[86]
Sweden	2014–2016	Feces	4/90 (4.4)	RT-PCR	[40]
		Lymph nodes (mesenteric)	6/67 (6.7)		
		Lymph nodes (sub-mandibular)	3/25 (12)		
		Tonsils	19/136 (14)		
Italy	2015–2018	Feces (not specified)	0/107	RT-PCR followed by culture (warm enrichment) for positive samples	[76]
Finland	2016	Blood (serum)	102/181 (56)	Enzyme-Linked Immunosorbent Assay (antibodies anti-*Yersinia* spp.)	[64]
		Organs (kidney, spleen)	22/130 (17)	RT-PCR followed by culture (warm enrichment) for positive samples	
Italy	2017	Muscle (shoulder area)	2/22 (9)	RT-PCR followed by culture (cold enrichment) for positive samples	[87]
Italy	2017–2019	Feces	19/305 (6.2)	Culture (warm and cold enrichment)	[67]
		Lymph nodes (mesenteric)	10/305 (3.3)		
Italy	2018–2020	Feces (rectal swabs)	54/287 (18.8)	Culture (cold enrichment)	[45]
Italy	2019	Carcass surface	12/36 (33.3)	Culture (warm enrichment)	[66]
		Tonsils	9/36 (25)		
		Meat (forearm area)	10/36 (27)		
Italy	2020	Lymph nodes (mesenteric)	0/64	Culture (cold enrichment)	[79]
		Carcass surface	0/64		
Italy	2020–2022	Feces (colon content)	18/66 (27.3)	Culture (cold enrichment)	[13]
		Lymph nodes (mesenteric)	3/66 (4.5)		
		Carcass surface	3/49 (6.1)		

RT-PCR: Real-Time Polymerase Chain Reaction.

### 3.3. Campylobacter spp. Occurrence and Epidemiological Insights

*Campylobacter* spp. is recognized as a major global public health concern accounting for 58.9% of all the reported and confirmed cases of zoonotic diseases in 2023 [30] with the main causative species of campylobacteriosis being *C. jejuni* and C. *coli*, which are known to have their primary reservoir in avian species, particularly chickens [88], but have also been isolated from wild animals, along with other species as *C. lari* and *C. lanienae*.

Table 3 summarizes the prevalence of *Campylobacter* spp. detected in different sample types (feces, lymph nodes, carcass surface, tonsils, organs, meat, bile) collected from wild boars in various European Union countries. The sampling period is indicated. Prevalence is expressed as the number of positive samples out of the total number examined, with the corresponding percentage reported in brackets. When available, *Campylobacter* identified species were reported.

In feces samples *Campylobacter* spp. prevalence can be comprised in a range between 0 and 66% [13,33,35,41,89,90]. This huge variability can be due to the fact that most studies apply methods that favor the growth of thermotolerant species (as *C. jejuni* and *C. coli*), thus excluding the possibility to detect also non-thermotolerant campylobacters, such as *C. lanianae*, *C. hyointestinalis* and *C. fetus* and underestimating the real prevalence [14]. Particularly *C. lanianae*, recognized as potential causes of human illness, is predominantly isolated from wild boars, domestic pigs and feral swine with prevalence which can be as high as 70% in Spain [13,89] and 40.8% in Italy [14,90]. However, many authors have also isolated *C. coli* with high prevalence rates. This is the case, for example, of the investigation of Castillo-Contreras et al. [13] who made a comparison of wild boar carriage of *Campylobacter* spp. in three different zones of the metropolitan area of Barcelona, with three different degrees of urbanization. Although no clear relationship was found between urbanization and specific *Campylobacter* species, *C. lanienae* was more frequently isolated in the less urbanized area, suggesting a natural diet as a potential source of infection. Conversely, an anthropogenic source of *C. coli* infection (e.g., exposure to rubbish) could be hypothesized, as its prevalence was twice as high in more urbanized areas compared to the less urbanized one. This pattern implies that increasing urbanization may shift wild boar exposure towards human-associated sources of contamination (e.g., refuse, wastewater, food scraps), thereby influencing both the composition of their enteric microbiota and their potential to act as vectors of pathogens with relevance for public health.

Some authors investigated the effect of potential influencing factors on *Campylobacter* spp. in wild boars: sex, weight, age did not reveal a significant impact on both organs and carcasses. *Campylobacter* spp. prevalence on wild boar carcasses was higher, although not significant, with environmental temperature on the day of hunting above 15 °C [41]. In the study by Carbonero et al. [89], the presence of artificial waterholes was significantly associated with an increased prevalence of *Campylobacter* spp. in wild boars. This is likely due to the ability of certain *Campylobacter* species to survive in aquatic environments, with outbreaks in humans linked to contaminated water having also been reported [91,92].

According to the same authors, *Campylobacter* spp. occurrence was significantly higher during the winter months. This may be due to the fact that lower temperatures support the bacterium’s ability to survive in the environment [89]. Moreover, experimental studies have shown that *Campylobacter* spp. can persist in water for longer periods at low temperatures, which could explain the observed association between waterholes and its presence in wild boars [93].

**Table 3 antibiotics-14-01246-t003:** *Campylobacter* spp. prevalence expressed as positive/total (%) in different sample types collected from wild boars in various countries of the European Union.

Country	Sampling Period	Source	Pos/Tot (%)	*Campylobacter* Species Identified (%)	Ref.
Germany	2006–2007	Muscle (various carcass area)	3/127 (2.4)	*C. coli* (66.6), *C. jejuni* (33.3)	[94]
Spain	2009–2011	Feces (rectal content)	188/287 (66)	*C. spp.* (one C. jejuni)	[37]
Spain	2010–2011	Feces (rectal content)	10/41(24.4)	*C. coli* (21.1), other thermophilic species (79.9)	[35]
Spain	2011–2012	Feces (intestine or rectum)	49/126 (38.9)	*C. lanianae* (69.4), *C coli* (16.3), *C. jejuni* (4.11), others (10.2)	[89]
Italy	2012–2019	Feces (not specified) Liver Muscle	Not reported	*C. coli* (91.7)*, C. jejuni* (8.3)	[95]
Italy	2016	Feces (caecum content)	29/56 (51.8)	Not investigated	[41]
		Lymph nodes (mesenteric)	0/56		
		Carcass surface	5/30 (16.7)		
Spain	2015–2016	Feces (rectal swabs)	79/130 (60.8)	*C. lanienae* (46.2), *C. coli* (16.2), *C. hyointestinalis* (0.8)	[13]
Italy	2018–2019	Feces (rectal swabs)	78/183 (42.6)	*C. coli* (48), *C. lanianae* (42), *C. jejuni* (6), *C. hyointestinalis* (4)	[90]
		Carcass surface	10/55 (18.2)		
		Liver	9/187 (4.81)		
		Bile	3/152 (1.9)		
Italy	2019	Feces (rectal content)	38/76 (50)	*C. lanienae* (40.8), *C. hyointestinalis* (14.5), *C. coli* (7.9), *C. jejuni* (1.3), *C. fetus* (1.3)	[14]
Italy	2019	Carcass surface	4/36 (11.1)	*C. coli* (33), *C. jejuni* (33), other species (33)	[66]
		Meat (leg)	2/36 (5.5)		
Italy	2019–2020	Meat (leg)	0/28		[87]

### 3.4. Prevalence and Diversity of STEC

The pathogenic *E. coli* pathotypes include enteroaggregative *E. coli* (EAEC), enteropathogenic *E. coli* (EPEC) and Shiga toxin-producing *E. coli* (STEC), and its subgroup of enterohemorrhagic *E. coli* (EHEC) [12]. STEC are of particular concern for public health and STEC infections were the third cause of zoonoses with over 10,217 cases in humans in 2023 [30]. Their pathogenicity is primarily linked to the production of Shiga toxins (Stx1, Stx2 and variants). In humans, STEC infections can result not only in acute gastroenteritis but also in severe systemic complications with the most serious outcome being the hemolytic uremic syndrome (HUS), a condition defined by hemolytic anemia, thrombocytopenia, and acute renal failure [96]. The best-known STEC serotype is O157:H7, which has been associated with numerous large-scale outbreaks worldwide. However, non-O157 serotypes such as O26, O103, O111, and O145 are recognized as significant causes of disease [97].

STEC transmission mainly occurs via food and water contaminated with fecal material [98]. While many domestic and wild animals act as asymptomatic carriers due to the absence of Shiga toxins receptors, wildlife—particularly ruminants such as deer—represents an important reservoir contributing to environmental contamination and the spread of infection [99].

Also, wild boars have been described as carriers of *E. coli* O157:H7 and other non-0157 STEC strains that are potential human pathogens.

Table 4 summarizes the prevalence of STEC detected in different sample types (feces, lymph nodes, tonsils, carcass surface, meat, and meat products) collected from wild boars in various European Union countries. The sampling period is indicated. Prevalence is expressed as the number of positive samples out of the total number examined, with the corresponding percentage reported in brackets. When available, STEC serogroups (reported as non-O157 and O157:H7) and other identified *E. coli* pathotypes are also indicated, with the corresponding percentages provided in brackets.

Prevalence in feces samples of non-O157 STEC can vary between 3 and >28% [12,37,47,100,101,102,103,104].

In some investigations, lower prevalence, or even no STEC detection, could be influenced by the fact that only O157:H7 presence is investigated [105,106]. Although most of the investigations mostly identified non-O157 isolates, some authors found serogroups associated with clinical case. Dias et al. [103] identified O27 serogroup, associated with hospitalization with neonatal HUS and bloody diarrhea [97]. In a study conducted in Italy [107], beside a quite high STEC prevalence (21.7%) in rectal swabs, also a 6.3% of EHEC was identified, harboring the virulence gene *eae*, responsible for the typical attaching and effacing lesions and associated with EPEC. In Spain, Mora et al. [100] identified 0.38% of O157:H7 isolates from wild boars and other serotypes associated with HUS and human outbreaks, as O126:H11 and O128:H2, as well as serotypes associated with diarrhea as O146.

Other pathotypes are often identified from wild boar samples as EPEC [101,102,108], EHEC [107], but also Extra-intestinal pathogenic *E. coli* as uropathogenic [108].

**Table 4 antibiotics-14-01246-t004:** STEC prevalence expressed as positive/total (%) in different sample types collected from wild boars in various countries of the European Union. When possible, distinction between serotype O157:H7 and other serotypes (non-O157) has been reported.

Country	Sampling Period	Source	Pos/Tot (%)	STEC Serogroups (%) and Others Pathotypes Identified (%)	Ref.
Portugal	2009–2010	Feces (rectal swabs)	22/262 (8.4)	non-O157 (8.4), O157:H7 (0.38)	[100]
Spain	2009–2011	Frozen meat	1/36 (2.7)	non-O157 (100%)	[109]
		Meat products	2/21 (9.5)		
Spain	2009–2011	Feces (rectal content)	4/117 (3.4)	Detection method only for O157:H7	[106]
Sweden	2010–2011	Feces (not specified)	0/88	Detection method only for O157:H7	[105]
		Lymph nodes (mesenteric)	0/56		
		Tonsils	0/175		
Spain	2009–2011	Feces (rectal content) Carcass surface	11/301 (4) 12/310 (4)	non-O157 (100%)	[37]
Portugal	2013–2014	Feces (rectal content)	1/21 (4.8)	non-O157 (100%)	[103]
Spain	2013–2015	Feces (not specified)	3/90 (3.3)	non O157 (100%) *E. coli* EPEC (3.3)	[101]
Germany	2016	Feces (not specified)	37/536 (6.9)	O157:H7 (8.3) non-O157 (91.7)	[47]
Poland	2017–2018	Feces (rectal swabs)	64/152 (28.3)	non-O157 (92.1), O157 (7.9) *E. coli* EPEC (17.11)	[102]
Portugal	2017–2019	Feces (environment or rectal content)	8/56 (14)	non-O157 (100%)	[12]
Italy	2018–2019	Feces (rectal swabs)	13/200 (6.5)	Serogroups not reported EHEC (6.3), EAEC (5.7), aEPEC (3.4), and unspecific pathotypes also identified	[107]
Italy	2019–2020	Meat (forearm area)	12/28 (42.8)	Serogroups not reported	[66]
Switzerland	2021	Meat (not specified)	3/25 (12)	non O157 (100%)	[110]
Switzerland	2022–2023	Feces (colon content)	13/59 (22)	non O157 (100%)	[104]

STEC: Shiga toxin-producing *E. coli*; EPEC: enteropathogenic *E. coli*; EHEC: enterohemorrhagic; EAEC. enteroaggregative *E. coli*; aEPEC: atypical enteropathogenic *E. coli.*

### 3.5. Wild Boar as Carriers of Enteric Pathogens

Carriers were defined as animals testing positive in feces and/or lymph nodes and/or tonsils. Their relevance lies in the fact that infected wild boars may contaminate the environment through fecal dispersion, carcass or internal organs during dressing operations, or transmit pathogens directly to humans or to the environment during handling. This aspect is particularly critical when considered within a One Health perspective, as carriers may represent a bridge for pathogen dissemination across wildlife, domestic animals, humans, and environmental compartments.

Appendix A illustrate the prevalence of carrier wild boars reported in different European Union countries. As previously noted, these values must be interpreted with caution due to differences in diagnostic methods, sampling strategies, and analytical approaches, which may influence the outcomes.

Overall, the prevalence of carrier wild boars testing positive in feces and/or lymph nodes and/or tonsils ranges from approximately 0 to 24.8% for *Salmonella* spp. [11,13,33,34,35,36,37,38,39,40,41,42,43,44,45,46,47,49,50,66,67], and from 8.2 to 33.3% for *Y. enterocolitica* [13,33,40,45,66,67,70,74,75,76,78,79]. For *Campylobacter* spp. and STEC, carrier prevalence has been assessed at the fecal level only and ranges from 24.4 to 66% [13,14,35,36,37,41,89,90,95] and 3.3 to 28.3%, respectively [12,37,47,100,101,102,103,104,105,106,107].

Several authors investigated more than one pathogen simultaneously, and in some cases, co-infections were reported. For example, Cilia et al. [45] detected both *Salmonella enterica* subs. *houtenae* and *Y. enterocolitica* in the feces of one wild boar. Sanno et al. [40] reported two animals positive for both *Salmonella* spp. and *Y. enterocolitica* in the tonsils, whereas Siddi et al. [11] identified two wild boars positive for both pathogens in feces or lymph nodes. Other studies screened for multiple pathogens (e.g., *Salmonella*, *Campylobacter*, STEC, *Y. enterocolitica*) but did not report co-infections [13,35,37,41].

The identification of carrier wild boars underscores their potential role in food safety hazards and AMR dissemination, reinforcing the need to consider this species as an integral component of One Health surveillance programs.

### 3.6. Focus on the Risk of Meat Contamination; Presence of Zoonotic Pathogens on Carcass Surface, Organs, and Meat

Several factors influence the microbiological quality of game meat, including the microorganisms carried by the animal, hunting conditions, and practices during butchering, handling, and storage [65]. Consequently, the safety of wild boar meat depends both on the health status of the hunted animal and on the hygiene conditions of slaughtering and processing environments. These aspects may lead to meat contamination and pose a risk of zoonotic infection to consumers [111]. For this reason, it is particularly important to assess the presence of enteric zoonotic pathogens on the carcass surface and, when relevant, in muscles and edible organs such as the liver. Nevertheless, few studies include these matrices in their investigations.

*Salmonella* spp. has been detected on carcasses of hunted wild boars with low prevalence rates, ranging between 0 and approximately 2.5%, when evaluated using swab or sponge methods according to ISO 17604 or comparable techniques [11,37,41,46,50,63,66,67], either after skinning [49] or through the excision method [65].

*Y. enterocolitica* has been detected with prevalence rates ranging from 0 up to over 30% on carcass surfaces, and although only biotype 1A is usually identified, these results highlight a potential risk for consumers [11,66,83]. When investigated at muscle level, *Y. enterocolitica* has been detected with variable prevalence: 0% in *Longissimus dorsi* [86], 9% in shoulder samples [87], 14.8% in diaphragm and leg [61], and 27% in the forearm area [66].

In northern Italy (Liguria), Modesto et al. [85] investigated *Y. enterocolitica* in 4890 wild boar liver samples, reporting a prevalence of 2.6%. Most isolates (92.9%) were identified as biotype 1A, 3% as biotype 1B, and <1% as biotype 2. Notably, among the 1B isolates, two belonged to serotype O:8, one of the most frequent in human yersiniosis, suggesting a possible anthropogenic origin.

Liver samples have also been investigated for the presence of other pathogens, such as *Salmonella* spp.—with reported prevalence rates ranging from 1.7% to 6% [45,49,62]—and *Campylobacter* [95], in which the most common species involved in human campylobacteriosis (*C. coli* and *C. jejuni*) were detected, although prevalence data were not reported in this case.

The isolation of potentially pathogenic *Y. enterocolitica* strains and other enteric pathogens from wild boar liver is particularly relevant, as this organ is traditionally consumed only lightly cooked or even used raw in the preparation of certain meat products in Italy [62].

*Campylobacter* spp. was detected in 16.7% of wild boar carcasses in northern Italy [41]. In that study, the authors observed a correlation between bacterial load—expressed as total viable count (TVC) and *Enterobacteriaceae*—and the presence of *Campylobacter* spp. on carcasses, with prevalence increasing from 16.7% to 23% and 25% at TVC levels of 3–4 and >4 log CFU/cm^2^, respectively. Other studies reported prevalence ranging between 2% and 18% [66,90,94], with *C. coli*, *C. jejuni*, and *C. lanienae* being the most frequently identified species.

As for STEC, prevalence rates of 12% in meat samples from the forearm area [87], 9.5% in meat products and 2.7% in frozen meat samples have been reported [109]. Some isolates, although non-O157, belonged to serotypes and carried virulence genes associated with human infections, thus representing a potential public health concern.

More rarely, the presence of enteric pathogens has been investigated in other organs (e.g., kidneys, spleen) or in different types of samples such as blood and meat juice extracted from diaphragm, with highly variable prevalence results, mostly depending on the analytical method applied–as real-time PCR, Enzyme-Linked Immunosorbent Assay (ELISA)-rather than culture-based methods [64,84].

Overall, these results highlight the need for continuous monitoring and for the implementation of Good Hygiene Practices (GHP) and Good Manufacturing Practices (GMP) throughout wildlife hunting, field dressing, and carcass processing. Such measures are essential to minimize contamination risks and ensure the safety of wild boar meat, which may otherwise serve as a potential vehicle for zoonotic pathogens.

Table 5 summarizes the prevalence ranges of *Salmonella* spp., *Y. enterocolitica*, STEC, and *Campylobacter* spp., expressed as minimum–maximum percentages, detected in different sample types collected from wild boars across various European Union countries. These ranges reflect the substantial heterogeneity among the included studies, which differ in sampling design, matrices, and analytical methodologies (cultural, molecular, or immunoenzymatic). As a result, the reported values should be interpreted as broad indicative intervals rather than directly comparable prevalence estimates.

## 4. Antimicrobial Resistance Profiles of Enteric Pathogens Detected from Wild Boars in European Union

Despite a vast and growing body of literature on AMR in medical and veterinary contexts, research addressing the complex transmission dynamics of AMR in environmental and wildlife compartments remains scarce, although such investigations are essential to fully operationalize the One Health approach [112].

In theory, wildlife animals are not exposed to antibiotic treatments; however, their direct and indirect interactions with livestock, domestic animals, and human-influenced environments, combined with their ability to move freely across diverse habitats, increase their likelihood of encountering selective agents, commensal bacteria, and resistant microorganisms [5]. Such interactions are thought to facilitate adaptive processes in commensal bacteria and promote the horizontal transfer of resistance genes within wildlife bacterial communities [113].

Wild boars, in particular, are increasingly recognized as potential sources of resistant foodborne pathogens for humans, primarily through the handling and consumption of their meat [114,115].

From the analysis of the scientific literature, it emerges that the prevalence rates of antibiotic resistance, as well as the phenotypic and genotypic resistance profiles found in enteric pathogens isolated from wild boars in various European Union countries, show considerable variability depending on the geographical location. Several factors contribute to this heterogeneity. Differences in environmental contamination levels, livestock density, and farming practices strongly influence the exposure of wild boars to resistant bacteria and selective agents [5,47,113]. Moreover, variations in sampling strategies (e.g., target organs, sample size, and season of collection), bacterial species investigated, and laboratory methodologies—most commonly the Kirby–Bauer disk diffusion test or the broth microdilution method (MIC)—further affect the comparability of results across studies [5,113]. Finally, the ecological behavior and movement patterns of wild boar populations, which differ markedly across European regions, may also contribute to the spatial heterogeneity of AMR occurrence in this species [112].

A useful approach adopted in recent studies is the interpretation of phenotypic AMR profiles based on the epidemiological cut-off values (ECOFFs), which allow the distinction between wild-type (WT) and non-wild-type (NWT) strains with acquired resistance [116,117]. This provides essential information to facilitate comparison between studies, to understand AMR dissemination dynamics and to assess the potential threat it may pose to public health.

In the analyzed literature, strains are defined as multidrug-resistant (MDR) when they are resistant to at least one antimicrobial in three or more classes, while they are defined as extensively drug-resistant (XDR) when they are resistant to at least one antimicrobial in all but one or two of the tested classes [118].

Figure 3 report the number of resistant *Salmonella* spp., *Y. enterocolitica*, *Campylobacter* spp. and STEC detected from wild boars referred to antibiotic classes.

In Figure 4, a heat map illustrating and comparing the antibiotic resistance profiles of *Salmonella* spp., *Yersinia enterocolitica*, STEC, and *Campylobacter* spp. isolated from wild boars is showed. The heat map displays the percentage of resistant isolates for each bacterial species across a panel of tested antibiotics. Colors range from yellow (low resistance) to dark red (high resistance), allowing visual identification of critical resistance patterns. Each row corresponds to an antibiotic, while each column represents a bacterial group, enabling side-by-side comparison of their resistance frequencies.

It is well known that antibiotics used in veterinary medicine and agriculture often belong to the same classes as those employed in human medicine [119]. Among enteric pathogen isolated from wild boars, resistant strains have also been identified to antibiotic classes categorized as critically important, as Enterobacterales resistant to carbapenems and third-generation cephalosporins, and highly important, as non-typhoidal *Salmonella* spp. resistant to fluoroquinolones [62].

### 4.1. AMR Patterns in Salmonella Isolates

As regards *Salmonella* spp., high prevalence rates of resistance to sulfonamides have been reported. Resistance rates exceeding 85% for sulfamethoxazole were found in Italy [38] and in Spain [120]. Similarly, Razzuoli et al. [62] reported a resistance prevalence rate of 96% against triple-sulfa compounds. High resistance levels have also been observed for streptomycin and tetracycline with prevalence rates of 41.7% for both antibiotics in Italy [42] and of 46.2% and 25.3% in Spain [120] for streptomycin and tetracycline, respectively. Notably, even higher streptomycin resistance (61.1%) was detected by Cilia et al. [45].

#### AMR in *Salmonella* Isolates Against Critically Important and High Important Antibiotics

A particularly concerning finding is the detection of resistance to colistin in Italy and Spain [38,47,62]. This result is of particular concern because colistin represents one of the few remaining therapeutic options against multidrug-resistant Gram-negative bacteria, and the emergence of plasmid-mediated resistance genes (such as *mcr* variants) facilitates their rapid spread across bacterial species and ecosystems [121]. Piras et al. [46] identified *Salmonella* spp. strains resistant to fosfomycin, along with the presence of the corresponding resistance gene *FosA7*. Fosfomycin may represent a valuable therapeutic option for the treatment of cystitis caused by extended-spectrum β-lactamase (ESBL)-producing *Enterobacteriaceae* [122]. Notably, this resistance was detected in *Salmonella* spp. strains belonging to the serovars Agona and Derby, which were among the fifteen most frequently reported serovars in human salmonellosis cases in Europe in 2023 [30]. In the study by Zottola et al. [38], in addition to the high resistance rates reported, 54% of the isolates were classified as MDR. Alarmingly, resistance was also detected—albeit at low levels—against third-generation cephalosporins (ceftiofur and cefotaxime), which are classified by the World Health Organization (WHO) as critically important antimicrobials. Similar resistance trends were observed by Razzuoli et al. [62] and Gil-Molino et al. [120], who additionally observed resistance to fluoroquinolones, which belong to the WHO’s high-priority group for antimicrobial resistance.

### 4.2. AMR Patterns in Y. enterocolitica Isolates

As regards *Y. enterocolitica*, high rates of resistance to β-lactam antibiotics, such as ampicillin and first- and second-generation cephalosporins, due to β-lactamase production, are well documented in the literature and recognized by both the Clinical and Laboratory Standards Institute (CLSI) and EUCAST [123,124]. An interesting result emerged from the investigation by Modesto et al. [85], who observed an increase in resistance to sulfonamide compounds, particularly sulfisoxazole and triple-sulfa, between 2014 and 2017, reaching 35% of resistant strains for both drugs by the end of the observation period. Notably, the same study also reported resistance to ceftiofur in 7% of the isolates and a substantial increase in MDR strains from 9.5% to 40% over the same period. Furthermore, resistance to erythromycin has occasionally been reported not only in wild boars [125] but also in other wild species [76].

### 4.3. AMR Patterns in STEC Isolates

AMR is not often determined for STEC isolates, as antibiotic treatment of human infections caused by STEC can enhance the production of Shiga toxins, potentially worsening clinical outcomes [126,127]. When AMR is investigated in STEC isolates from wild animals, high levels of susceptibility are generally reported [76,109]. By applying epidemiological cut-off values (ECOFFs), Dias et al. [12] identified non-wild-type (NWT) phenotypes in STEC strains isolated from wild boar fecal samples for nitrofurantoin and sulfamethoxazole/trimethoprim (100%), imipenem (classified by WHO as critically important), and tobramycin (87.5%), as well as for amikacin (37.5%) and chloramphenicol (25%). Interestingly, full susceptibility had been reported for the same isolates using the disk diffusion method. The same study also found a high proportion (87.5%) of multidrug-resistant (MDR) strains. Similarly, a previous investigation conducted in Germany identified 4% of NWT STEC isolates from wild boars showing resistance to compounds belonging to seven different antimicrobial classes, including fluoroquinolones and β-lactams [47].

### 4.4. AMR Patterns in Campylobacter spp. Isolates

Few studies have investigated AMR in *Campylobacter* spp. isolates from wild boars. Carbonero et al. [89] in Spain examined several wild artiodactyl species, including wild boars and reported high levels of resistance to erythromycin (95.2%), followed by ciprofloxacin (62.5%), tetracycline (47.1%), and streptomycin (45%), but results were not differentiated between the animal species. Similar resistance trends were observed in a more recent study by Castillo-Contreras et al. [13], who detected high resistance rates to tetracycline and ciprofloxacin (both 66%) and to streptomycin (43%). Additionally, they reported very high resistance to nalidixic acid (95%), with none of the isolates being pansusceptible, and 67% of *C. coli* strains classified as MDR. The AMR profiles observed in *C. coli*, along with their detection in areas characterized by a high or medium degree of urbanization, suggest an anthropogenic origin for these resistant strains. Moreover, resistance to ciprofloxacin is particularly alarming, as it is one of the two antimicrobials considered critically important for the treatment of human campylobacteriosis [128].

## 5. Future Perspective in AMR: Studying the Fecal Resistome

Studies on antimicrobial resistance (AMR) in wild animals often face several limitations: (1) variability in methods used across studies; (2) relatively small sample sizes; (3) experimental designs driven by sampling convenience and short timeframes; (4) limited numbers of wild species investigated, with wild boars being the most frequently studied; and (5) predominantly descriptive approaches that provide only a snapshot of AMR occurrence. These limitations result in poor spatial and temporal coverage of AMR data and leave significant gaps in understanding the mechanisms of AMR acquisition and transmission routes [25,129].

A promising approach was proposed by Dias et al. [12], who examined the presence and diversity of antibiotic resistance genes (ARGs) in wild boar fecal microbiomes using metagenomic and culture-based methods (high-throughput qPCR and phenotypic testing of *E. coli* and *Enterococcus* spp.). In 56 samples from three Portuguese areas with different levels of human impact, they identified 62 ARGs associated with nine antibiotic classes—mainly tetracyclines and aminoglycosides—and 20 mobile genetic elements, including integrons. Regions with greater anthropogenic pressure exhibited higher ARG diversity and abundance, highlighting the influence of human activity.

Phenotypic resistance testing of *E. coli* and *Enterococcus* spp. isolates supported the molecular findings. Overall, 27% of *E. coli* and 83% of *Enterococcus* spp. were resistant to at least one antibiotic, primarily β-lactams, aminoglycosides, macrolides, and tetracyclines, with relatively few multidrug-resistant strains. When evaluated using epidemiological cut-off values (ECOFFs), 45% of *E. coli* and 38% of *Enterococcus* spp. were classified as non-wild-type (NWT), indicating acquired resistance beyond the natural susceptibility range, while the remaining strains were wild-type (WT), displaying only intrinsic resistance. The proportion of NWT strains was higher in areas with stronger human influence.

These results suggest that wild boars can serve as ecological sentinels for monitoring AMR in the environment. The study underscores the importance of including wildlife in national AMR surveillance programs and highlights the value of a One Health approach to safeguard both public and environmental health.

## 6. From Forest to City: The Wild Boar as an Example of Synurbization

The rise in wild boar populations and their expansion, driven by the previously mentioned factors, has been accompanied by synurbization—a specific form of synanthropization that refers to the adaptation of animal populations to human-modified environments [130,131]. This process fosters new interactions between wild boars and humans, often leading to conflicts, potential attacks on pets or people, traffic accidents, and increased zoonotic risks [132].

The Barcelona Metropolitan Area (MAB) represents an exemplary One Health setting, with one of the most thoroughly documented cases of wild boar synurbization. Population densities ranged from 5–15 individuals per km^2^ between 2004 and 2022 [133,134,135]. The area encompasses a gradient of urbanization: the city of Barcelona and the Universitat Autònoma de Barcelona (UAB) are predominantly urban, though the UAB retains more gardens, forests, and agricultural patches (60%) than the city (28%). In contrast, the nearby Serra de Collserola Natural Park (CNP) consists mainly of natural habitats, including scrubland, forest, and grassland, interspersed with recreational spaces and built-up areas [136]. This habitat mosaic provides a unique context to study how wild boars adapt to varying degrees of urbanization.

As previously mentioned, Castillo-Contreras et al. [13] examined the occurrence of zoonotic *Campylobacter* spp. in wild boar feces from the MAB, aiming to assess the genetic diversity of isolates, their possible link with anthropogenic sources, and the relationship between *Campylobacter* spp. carriage and the level of urbanization. Among the species identified, *C. lanienae* showed a prevalence of 46%, being more common in wild boars from less urbanized areas. This suggests that infection is likely associated with a diet based on natural resources rather than human-derived sources. Conversely, the higher prevalence of *C. coli* observed in the two most urbanized areas—approximately double that found in the less urbanized zone—points to a probable anthropogenic origin of *C. coli* infections. Similarly, Darwich et al. [137] reported *E. coli* strains carrying critical β-lactam resistance genes in the MAB. Wild boars foraging in urban and peri-urban areas were found to be more frequently exposed to AMR *E. coli* than those inhabiting natural environments.

Interesting findings have also emerged from studies on microorganisms other than enteric pathogens, such as *Enterococcus* spp. Navarro-Gonzalez et al. [114] compared two wild boar populations: one inhabiting Collserola Natural Park and its surroundings (classified as urban wild boars) and another from the National Game Reserve Ports de Tortosa i Beseit, a remote rural area. The authors reported a significantly higher prevalence of *E. faecium* resistant to tetracycline and showing high-level streptomycin resistance in urban wild boars compared with their rural counterparts. The meaning of these results lies in the fact that both streptomycin and tetracycline are among the most commonly used antibiotics in livestock production, indicating that urban wild boars may acquire resistant bacteria through contact with environments contaminated by agricultural runoff, manure, or other human-related sources. In an earlier study, the same authors [35] also detected, in an urban wild boar from Collserola Natural Park, a linezolid-resistant *E. faecalis* strain—strongly suggesting an anthropogenic origin, since linezolid is a fully synthetic antibiotic used exclusively in human medicine, and resistance remains extremely rare in isolates from food-producing animals [138,139].

These studies provide a valuable One Health model, illustrating how the interaction between wildlife, human activities, and the environment shapes pathogen circulation and antimicrobial resistance dynamics in synurbized wild boar populations.

## 7. Conclusions and Future Perspective

The growing evidence on enteric pathogens and antimicrobial resistance (AMR) in wild boars highlights their role as a sentinel species within the One Health framework. Current data reveal considerable variability in prevalence and resistance profiles across Europe, driven by ecological, anthropogenic, and methodological factors. While resistance to critically important antimicrobials remains relatively sporadic, its presence in wild boar isolates underscores the permeability of ecological boundaries between humans, livestock, and wildlife.

Despite the increasing recognition of wild boars as indicators of AMR spread, several knowledge gaps remain. Longitudinal studies are scarce, limiting our understanding of temporal trends and persistence of resistant strains in wildlife populations. Genomic and metagenomic analyses are still underutilized, hindering the identification of transmission pathways between wildlife, domestic animals, and humans. Furthermore, the impact of environmental contamination and the role of synurbic behavior in shaping AMR dynamics remain poorly quantified.

These findings call for the development of harmonized surveillance systems that integrate wildlife monitoring with existing AMR control programs for livestock and humans. Policy frameworks should promote standardized methodologies for sampling, detection, and reporting of AMR in wildlife. Additionally, cross-sectoral collaboration and data sharing are critical to inform risk assessment, management strategies, and public health interventions aimed at minimizing AMR dissemination across ecosystems.

Wild boars, given their expanding distribution and increasing interaction with human environments, represent not only a challenge for public health but also a unique opportunity to fill critical research gaps and guide evidence-based policy actions within a One Health approach.

## Figures and Tables

**Figure 1 antibiotics-14-01246-f001:**
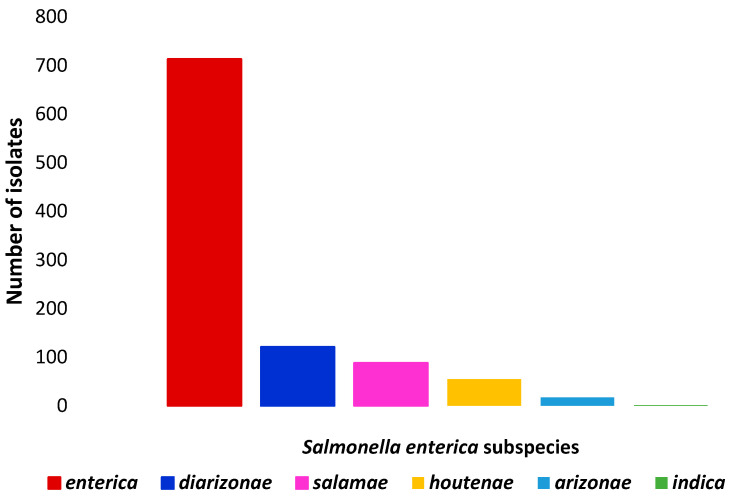
Number of isolates of each *Salmonella enterica* subspecies identified in wild boars.

**Figure 2 antibiotics-14-01246-f002:**
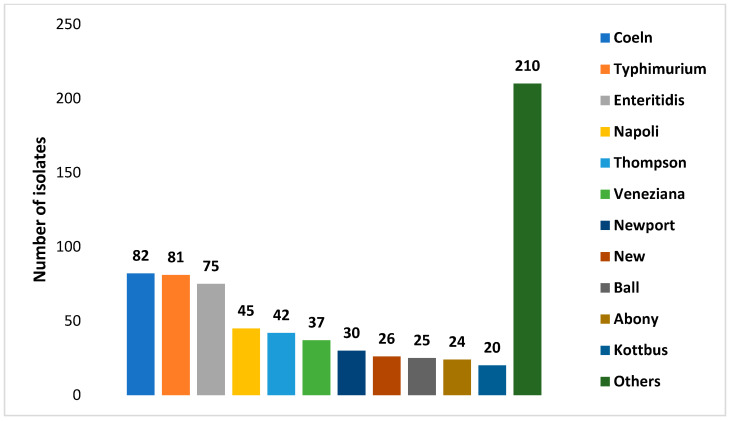
Numbers of isolates of each *Salmonella enterica* subs. *enterica* serotypes identified in wild boars’ isolates.

**Figure 3 antibiotics-14-01246-f003:**
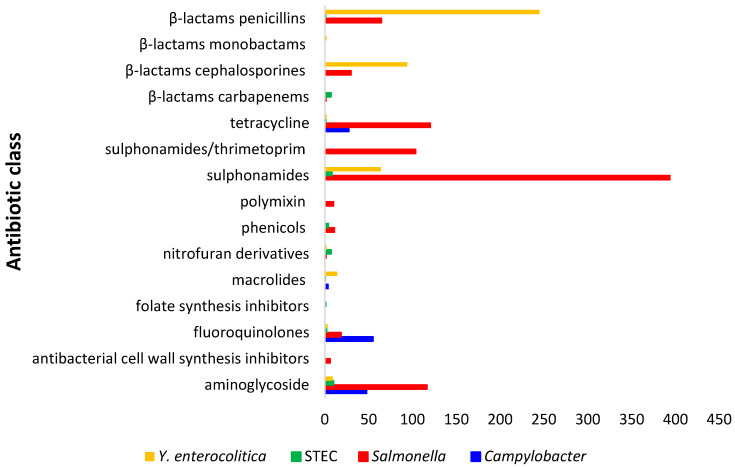
Number of isolates of *Salmonella* spp., *Y. enterocolitica*, STEC and *Campylobacter* spp. resistant to different antibiotic classes.

**Figure 4 antibiotics-14-01246-f004:**
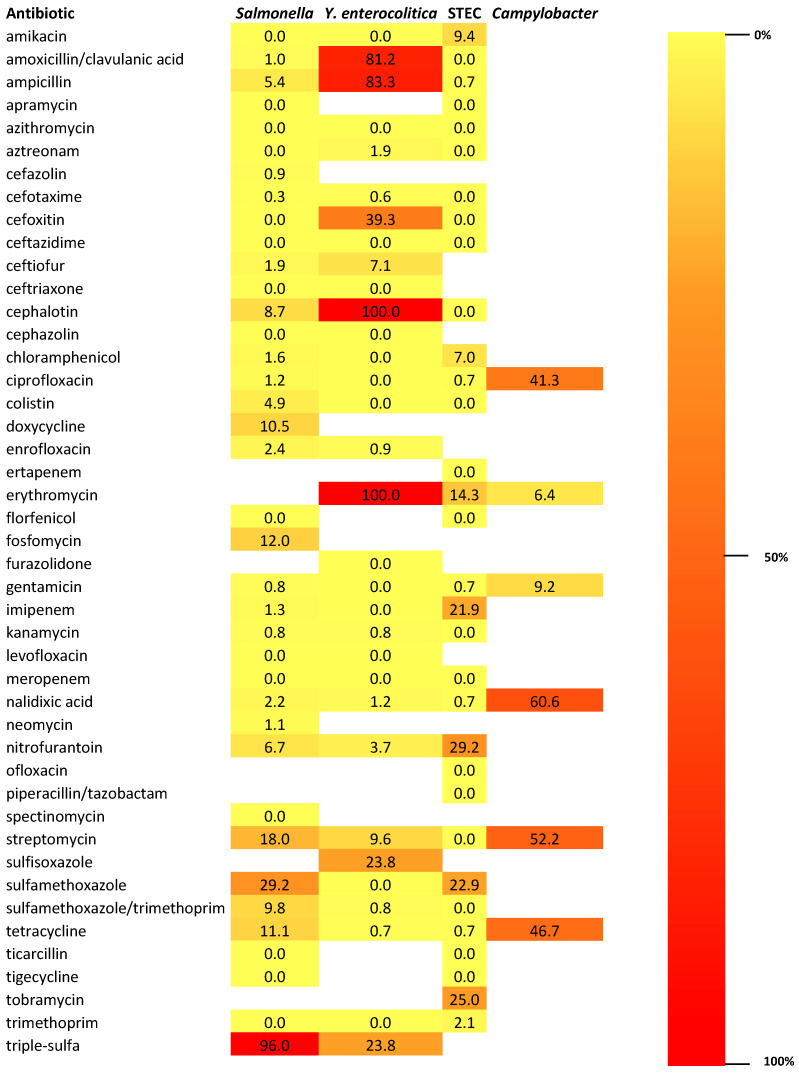
Heat map illustrating and comparing the antibiotic resistance profiles of *Salmonella* spp., *Yersinia enterocolitica*, STEC, and *Campylobacter* spp. isolated from wild boars. The sidebar represents the color scale used in the heat map, showing how colors correspond to resistance levels—from 0% (light yellow) to 100% (dark red). Triple-sulfa: combination of sulfadiazine, sulfamerazine, and sulfamethazine.

**Table 5 antibiotics-14-01246-t005:** Prevalence ranges of *Salmonella* spp., *Y. enterocolitica*, STEC, and *Campylobacter* spp., expressed as minimum–maximum percentages, detected in different sample types collected from wild boars across various European Union countries.

Source	*Salmonella* spp.	*Y. enterocolitica*	STEC	*Campylobacter* spp.
Feces	0–35.6	0–27.3	0–28.3	24.4–66
Lymph nodes	0–17.8	0–12	0	0
Tonsils	0–35.8	8–33.3	0	n.i.
Carcass surface	0–7.8	0–33.3	4	11.1–18.2
Muscle/meat	2.8–3.6	0–27	2.7–42.8	0–2.4
Liver	1.7–6	2.6	n.i.	4.81
Other organs (kidneys, spleen)	2–4.6	17	n.i.	n.i.

n.i.: not investigated.

## Data Availability

Not applicable.

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
