# Peer review of "Enteric Pathogens in Wild Boars Across the European Union: Prevalence and Antimicrobial Resistance Within a One Health Framework"

_antibiotics, 2025, doi:10.3390/antibiotics14121246_

Round 1
Reviewer 1 Report
Comments and Suggestions for Authors
Dear colleagues!
In general, the message that the authors tried to convey with this article is clear. People — animals — the environment are closely interconnected.
I ask you to make some corrections to the design of the material:
1. If you indicate Salmonella, you must indicate which one or Salmonella spp., if you do not identify it.
The same remark applies to other bacteria mentioned in the article.
2. Yersinia enterocolitica — after the first mention in the text → Y. enterocolitica
The same remark applies to other bacteria mentioned in the article.
3. Despite the fact that the article is a review, it must contain all structural components.
I did not see the “Materials and Methods” section in the text, which should indicate where and how you searched for information, what resources you used, over what period of time you analyzed the data, etc.
I remind you that using literature from 15-20 years ago will make your research results unrepresentative, unless you have tried to show the process in dynamics.
Sincerely,
Reviewer
Reviewer 2 Report
Comments and Suggestions for Authors
This manuscript provides a comprehensive review of enteric pathogens and antimicrobial resistance (AMR) in wild boars across the European Union within a One Health framework. It effectively compiles data on Salmonella, Yersinia entero colitica, Campylobacter spp., and Shiga toxin-producing E. coli, highlighting their epidemiological and public health relevance. While scientifically sound and well referenced, the paper is overly descriptive, with limited synthesis and repetition.
Abstract
“Wild boars, due to their widespread occurrence…” starting should be “Because of their wide distribution across natural, agricultural, and urban habitats, wild boars serve as ideal sentinels for AMR surveillance.”
The last three sentences are conceptually strong but verbose; merge and simplify to highlight implications and key gaps.
The objectives of the work are missing.
Introduction
Lines 35–45: Combine population growth and ecological adaptability into a single concise paragraph.
Add a brief note on the pathogen spillover by wild boars.
“showed a systematic increasing” should be “have shown a systematic increase.”
Lines 61–70: The discussion of AMR as a “silent pandemic” is compelling but overly rhetorical; one concise sentence would suffice.
Lines 71–83: The justification of wild boars as One Health sentinels is strong. Consider ending with a statement outlining the review’s objectives (e.g., “This review synthesizes available data on enteric pathogens and AMR in wild boars to assess their role in environmental transmission dynamics within the EU.”).
Section 2: Enteric Pathogens
Consider including an overall summary table comparing prevalence ranges among pathogens.
2.1 Salmonella (Lines 101–171)
Lines 115–126: Too method-specific; suggest condensing technical sampling details into one sentence.
Line 139–163: Age, sex, and seasonality factors are discussed well but could be summarized more concisely.
Ensure uniform use of italics for species names (e.g., Salmonella enterica).
2.2 Yersinia enterocolitica (Lines 198–260)
Add a short concluding remark connecting wild boar isolates to potential zoonotic implications.
Tables are clear but need captions explaining abbreviations (e.g., BT = biotype).
Bar diagram is recommended for Figure 1 & 2.
2.3 Campylobacter spp. (Lines 261–305)
Line 272: “most part of the studies apply methods” should be “most studies apply methods.”
Line 280-286: Clarify the implications of urbanization trends mentioned at lines 280–286.
2.4 STEC (Lines 307–350)
Typo: “recognises” should be “recognized.”
- Meat Contamination (Lines 355–406)
Integrate with Discussion or summarize to emphasize implications for food safety and wildlife monitoring. Example edit: “These findings underline the need for improved hygiene in game meat handling and highlight the potential role of wild boars in transmitting zoonotic pathogens through contaminated carcasses.”
Section 3: Antimicrobial Resistance (Lines 408–521)
Figures 3 and 4 are useful but should have clearer legends (define MDR, NWT, ECOFF).
In figure 3, what is pansusceptible ? It is not a class of antibiotics. Revise it.
- coli STEC means what ? write only Shiga toxin producing E. coli (STEC).
Figure 4, Gentamycin should be Gentamicin.
Name of antibiotics should be carefully spelled.
Lines 465–470: Create a subheading like “Critically Important Antimicrobials (CIA)” for better organization.
“comparable findings were reported” should be “similar resistance trends were observed.”
Ensure consistent use of WHO and EUCAST references.
Lines 540–548: Combine and shorten to avoid redundancy.
Define all abbreviations in figure legends.
Ensure consistent in-text referencing order (Figures 1–4 sequentially cited).
Comments on the Quality of English Language
The English is acceptable, but several typos were found and should be carefully corrected.
Reviewer 3 Report
Comments and Suggestions for Authors
34-83 Simplify and shorten excessively long sentences. Ensure that all cited references directly support the content, as the review currently includes excessive and overly descriptive material.
268 The "spp." abbreviations are never italicized.
339, 341, 346 “E. coli” - the species should be write with italic. Please revise this aspect in the entire manuscript.
Author Response
the comments performed by the referees to the manuscript " Enteric pathogens in wild boars across the European Union: prevalence and antimicrobial resistance within a One Health framework” have been addressed, and the paper has been revised accordingly. The tables below display the precise modifications based on the individual comments. Revisions in the text are highlighted with different colours (red for Reviewer 1, blue for Reviewer 2, purple for Reviewer 3 and green for Reviewer 4).
Reviewer 3
|
|
Modifications in the text (purple) |
|
34-83 Simplify and shorten excessively long sentences. Ensure that all cited references directly support the content, as the review currently includes excessive and overly descriptive material. |
The manuscript has been carefully revised also in order to avoid long sentences, particularly in the introduction section. We kindly ask to the referee, if him/her should identify any specific point that still require further elaboration, we will address them accordingly. |
|
268 The "spp." abbreviations are never italicized. |
Revised accordingly throughout the manuscript. |
|
339, 341, 346 “E. coli” - the species should be write with italic. Please revise this aspect in the entire manuscript. |
Revised accordingly throughout the manuscript. |
Reviewer 4 Report
Comments and Suggestions for Authors
Abstract: need to be shortened and more focused on the significance of the findings
Introduction: please state clearly the aim of the current review
It would be informative to provide a graph highlighting the link between wild boars, domestic animals, humans, and the environment in the context of One Health Concept and AMR dissemination.
It is important for such reviews to make sure that as many publications from each country is included not only one publication per country and add a meta summary for each table.
It would be more informative to visualize the geographic trends or distribution of enteric pathogens as well as their AMR-profile among Europe using a map.
Also, please clarify interpretation of mean resistance prevalence and strengthen the one health concept.
Conclusion and future perspective: highlight the research gaps and policy needs
Round 2
Reviewer 1 Report
Comments and Suggestions for Authors
Dear colleagues!
I fully support the authors in that such studies should be conducted and are important in the aspect of One Health.
Using a single database doesn't give your review much value, as the "picture" is not fully reflected. Wishes for the future - when writing articles, do not limit yourself to the SCOPUS database, as not all “interesting” materials are present there.
Sincerely,
Reviewer
Reviewer 2 Report
Comments and Suggestions for Authors
The manuscript is now scientifically sound and improved greatly.
Reviewer 4 Report
Comments and Suggestions for Authors
Accept